# Characterization of Dental Pulp Stem Cells Response to Bone Substitutes Biomaterials in Dentistry

**DOI:** 10.3390/polym14112223

**Published:** 2022-05-30

**Authors:** Rosanna Di Tinco, Ugo Consolo, Alessandra Pisciotta, Giulia Orlandi, Giulia Bertani, Milena Nasi, Jessika Bertacchini, Gianluca Carnevale

**Affiliations:** 1Department of Surgery, Medicine, Dentistry and Morphological Sciences with Interest in Transplant, Oncology and Regenerative Medicine, University of Modena and Reggio Emilia, 41125 Modena, Italy; rosanna.ditinco@unimore.it (R.D.T.); ugo.consolo@unimore.it (U.C.); alessandra.pisciotta@unimore.it (A.P.); giulia.orlandi@unimore.it (G.O.); giulia.bertani@unimore.it (G.B.); milena.nasi@unimore.it (M.N.); jessika.bertacchini@unimore.it (J.B.); 2Operative Unit of Dentistry and Maxillofacial Surgery, Department Integrated Activity-Specialist Surgeries, University-Hospital of Modena, 41125 Modena, Italy; 3CNR-Institute of Molecular Genetics “Luigi Luca Cavalli-Sforza”, Unit of Bologna, 40136 Bologna, Italy

**Keywords:** human dental pulp stem cells, neural crest, osteogenic differentiation

## Abstract

Bone substitute biomaterials (BSBs) represent a promising alternative to bone autografts, due to their biocompatibility, osteoconduction, slow resorption rates, and the ability to define and maintain volume for bone gain in dentistry. Many biomaterials are tailored to provide structural and biological support for bone regeneration, and allow the migration of bone-forming cells into the bone defect. Neural crest-derived stem cells isolated from human dental pulp (hDPSCs) represent a suitable stem cell source to study the biological effects of BSBs on osteoprogenitor cells involved in the physiological bone regenerative processes. This study aimed to evaluate how three different BSBs affect the stem cell properties, osteogenic differentiation, and inflammatory properties of hDPSCs. Our data highlight that BSBs do not alter cell proliferation and stemness markers expression, nor induce any inflammatory responses. Bone metabolism data show that hDPSCs exposed to the three BSBs distinctively secrete the factors supporting osteoblast activity and osteoclast activity. Our data indicate that (i) hDPSCs are a suitable stem cell source to study the effects of BSBs, and that (ii) the formulation of BSBs may condition the biological properties of stem cells, suggesting their versatile suitability to different dentistry applications.

## 1. Introduction

The loss of one or more dental elements leads to a vertical and horizontal atrophy of the alveolar bone, resulting in its significant three-dimensional resorption within the first six months. In order to place implants of the proper length, and to achieve a favorable crown-to-root ratio, a suitable bone volume is necessary [1,2,3]. Over the last few decades, efforts were made to find valid strategies aimed at favoring bone tissue regeneration through a patient-specific approach, such as bone autologous implants [4,5,6,7,8,9,10]. Although autogenous bone is still considered the most effective bone graft substitution material (“gold standard”) [11], several disadvantages were identified, including an additional intraoral or extraoral surgical site, resorption, morbidity, risk of bleeding, edema, and postoperative pain [12,13,14]. Therefore, there is an increasing interest in designing new biomaterials that support bone regeneration processes with a limited activation of the immune response and improve a stable implant interaction [15]. Among the available bone grafts, xenogeneic bone substitute biomaterials (BSBs) represent a promising alternative to autografts, due to their biocompatibility, osteoconduction, slow resorption rates, and the ability to define and maintain the volume for bone gain [16]. Most of the xenografts currently used have porcine and bovine origins, because of their similarity to the human bone, in terms of chemical composition (mainly carbonated hydroxyapatite and type I collagen) and structure [17]. Moreover, bone BSBs are composed of corticocancellous-collagenated bones, in the form of mixed granules of different size. These bone void fillers provide structural and biological support for bone regeneration and allow the migration of bone-forming cells into the bone defect.

Notwithstanding, while their clinical applications are well documented in literature [18,19,20], the downstream mechanisms of their functioning are not yet elucidated. Since the success of implants is primarily associated with their osseointegration, as well as their long-term permanence in function, it is important to understand the mechanisms that control the interface among BSBs, bone, and soft tissues around the implant site. In fact, regenerative dentistry is an emerging field, providing new challenges in modern dentistry, aimed at improving dental research and translating scientific knowledge into new future clinical treatments [21]. This approach is based on understanding the biological processes of healing and repair, which could be applied for enhancing the natural healing potential of the dental tissues or regenerating the damaged tissue [22].

To the best of our knowledge, this is the first study evaluating, from a biological point of view, the mechanistic effects of three BSBs on the surrounding microenvironment, by focusing on their biological effects on resident osteoprogenitor cells/ectomesenchymal stem cells involved in histointegration processes (hDPSCs). These stem cells have a peculiar origin from the neural crest and reside in the loose connective tissue entrapped in the pulp chamber of the teeth, where they preserve dental pulp homeostasis. Indeed, due to their origin, hDPSCs are able to differentiate between different lineages, including the osteogenic commitment [23]. In light of these properties, hDPSCs may provide a suitable cell source to assess the cell adhesion, viability, proliferation, and osteogenic differentiation when treated with BSBs.

## 2. Materials and Methods

### 2.1. Human DPSCs Isolation and Immune Selection

The study was carried out in accordance with the recommendations of the ethics committee of the province of Modena (Italy) (ref. number 3299/CE, 5 September 2017). Human DPSCs were isolated from third molars of adult subjects (*n* = 3; 30 to 35 years old) during routine extraction procedures, after obtaining their written informed consent in compliance with the Declaration of Helsinki. Cells were isolated from dental pulp, as previously described by Di Tinco et al. (2021) [24]. Briefly, after the extraction of the dental pulp, it was harvested from the teeth and underwent enzymatic digestion using a digestive solution (3 mg/mL type I collagenase, plus 4 mg/mL dispase in α-MEM). A cell suspension was first obtained by filtering pulp onto 100 μm Falcon cell strainers, plated in 25 cm^2^ culture flasks, and then expanded in standard culture medium (α-MEM supplemented with 10% heat-inactivated fetal bovine serum (FBS), 2 mM L-glutamine, 100 U/mL penicillin, and 100 μg/mL streptomycin; all from Sigma-Aldrich, St. Louis, MO, USA) at 37 °C, and 5% CO_2_. An immune selection was carried out on previously expanded hDPSCs using MACS^®^ separation kit (Miltenyi Biotec, Bergisch Gladbach, Germany), according to manufacturers’ instructions. Mouse IgM anti-STRO-1 and rabbit IgG anti-c-Kit (Santa Cruz, Dallas, TX, USA) were used as primary antibodies, and then revealed by magnetically labeled secondary antibodies: anti-mouse IgM and anti-rabbit IgG (Miltenyi Biotec). The immune selection allowed the isolation of a homogeneous hDPSCs population expressing STRO-1 and c-Kit. All the experiments were performed using STRO-1^+^/c-Kit^+^ hDPSCs.

### 2.2. Culture of hDPSCs in Presence of Bone Substitutes Biomaterials

Human DPSCs were cultured in presence of three BSBs, named OsteoBiol^®^ GTO^®^, OsteoBiol^®^ Gen-Os^®^, and OsteoBiol^®^ Apatos^®^, (Tecnoss^®^, Giaveno, Italy), in order to investigate their biological effects on hDPSCs. STRO-1^+^/c-Kit^+^ hDPSCs were seeded at a density of 3.0 × 10^3^ cell/cm^2,^ and after cell adhesion, BSBs were added to a culture medium at 10 mg/mL concentration. Human DPSCs were kept in culture for 48 h, and a morphological analysis was performed at three different time points (0, 24, 48 h) by phase-contrast light microscopy. At the end of the culture, BSBs were removed, together with the supernatants, and discarded, whereas the attached hDPSCs were processed, according to the subsequent experimental procedures. Human DPSCs cultured alone were used as control.

### 2.3. PBMCs Isolation and Co-Culture with hDPSCs

Human peripheral blood was collected from healthy donors (*n* = 5) who gave written informed consent, according to the guidelines of the ethics committee. Peripheral blood mononuclear cells (PBMCs) were isolated using Histopaque (Sigma-Aldrich, Saint Louis, MO, USA), according to the manufacturer’ instructions, and pre-activated as previously described [24]. In particular, direct co-culture systems were established by seeding hDPSCs at a cell density of 5000 cells/cm^2^, and cultured in RPMI 1640 medium supplemented with 10% FBS, 2 mM glutamine, 100 units/mL penicillin, and 100 mg/mL streptomycin. Upon stem cell adhesion, after 24 h, pre-activated PBMCs (aPBMCs) were seeded on stem cells at a 10:1 ratio, and kept in culture for 48 h. At the end of the co-culture, supernatants were collected for the subsequent experimental procedures. Human DPSCs and aPBMCs cultured alone were used as controls.

### 2.4. Evaluation of Cell Morphology and Stemness Markers in hDPSCs

After 48 h of culture, hDPSCs were fixed with ice-cold 4% paraformaldehyde in pH 7.4 phosphate buffer saline (PBS) for 15 min and washed in PBS. After rinsing with PBS, cells were subsequently permeabilized with 0.1% Triton X-100 in PBS for 5 min and blocked with 3% BSA in PBS for 30 min at room temperature. In order to perform a complete morphological characterization, after 48 h of culture in presence with the three BSBs, hDPSCs were stained with TRITC-conjugated, anti-phalloidin antibody (Abcam, Cambridge, UK), whereas a confocal immunofluorescence analysis was carried out in order to evaluate the expression of stemness markers in the same experimental groups. To this purpose, cells were incubated with the following primary antibodies: mouse IgM anti-STRO-1 and rabbit IgG anti-c-Kit, diluted 1:100 in 1% bovine serum albumin (BSA) in PBS (Santa Cruz, Dallas, TX, USA). Secondary antibodies (goat anti-mouse IgM Alexa488, goat anti-rabbit Alexa546) were diluted 1:200 (Thermo Fisher Scientific, Waltham, MA, USA). The multi-labelling immunofluorescence experiments were carried out, avoiding cross-reactions between primary and secondary antibodies. In any case, nuclei were stained with 1 μg/mL 4,6-diamidino-2-phenylindole (DAPI) in PBS for 5 min, and FluoroMount was used as anti-fading mounting medium (Sigma-Aldrich). Confocal imaging was performed using a Nikon A1 confocal laser scanning microscope. The confocal serial sections were processed with ImageJ software to obtain 3-dimensional projections, and image rendering was performed by Adobe Photoshop Software [24].

### 2.5. Cell Adhesion, Proliferation and Viability Bioluminescent Assays

Cell adhesion, proliferation, and viability were evaluated using real-time assays in hDPSCs cultured in the presence, or absence, of BSBs, using 96-well white plates (Corning, Kennebuck, ME, USA) and GloMax Discover Multimode Microplate Reader (Promega, Madison, WI, USA). In particular, cell adhesion and proliferation were measured with the RealTime-Glo^TM^ MT Cell Viability Assay (Promega), according to the manufacturer’s protocol. Shortly, after 24 h and 48 h of culture, cell culture medium was replaced with 50 μL of RealTime-Glo^TM^ reagent, diluted 1:1000 in standard culture medium, allowing the measurement of the viability of the attached stem cells. The luminescence was measured after 60 min incubation in a cell culture incubator (37 °C, 5% CO_2_). Cell viability was investigated using CytoTox-Glo^TM^ Cytotoxicity Assay (Promega), according to the manufacturer’s protocol. In brief, after 24 h and 48 h of culture, 50 μL of CytoTox-Glo^TM^ cytotoxicity assay reagent was added to all wells, and luminescence was measured after 15 min of incubation at room temperature. Then, 50 μL of Lysis buffer was added to all wells, and incubated again for 15 min at room temperature. The measured luminescent signal was associated with the total number of cells in each well. Viable cell luminescence was calculated by subtracting the luminescent signal of experimental cell death from total luminescent values. All the experiments were performed in triplicate.

### 2.6. Biological Effects of BSBs Treatment on hDPSCs

In order to investigate the biological effects of BSBs, the expression of α5/β1 integrins, lamin A/C, and osteopontin (OPN) was evaluated in hDPSCs after 48 h of culture with BSBs, using Western blot analysis performed on whole cell lysates (α5/β1 integrins, lamin A/C, and OPN) and supernatants (OPN). Briefly, 30 μg of protein extract per sample, quantified by a Bradford Protein Assay (Sigma Aldrich), or 50 μL of supernatants per sample, underwent SDS-polyacrylamide gel electrophoresis, and then was transferred to nitrocellulose membranes. The following primary antibodies were used: rabbit anti-integrin β1, rabbit anti-integrin α5 (Abcam), mouse anti-lamin A/C (Santa Cruz Biotechnology), mouse anti-OPN (Abcam) diluted 1:1000 in Tris-buffered saline Tween 20, with 2% BSA and 3% non-fat powder milk. Membranes were then incubated with HRP-conjugated anti-mouse and anti-rabbit secondary antibodies (Thermo Fisher Scientific) diluted 1:2000, for 1 h at room temperature. The immunoblotting was revealed using Clarity™ Western ECL substrate (Bio-Rad Laboratories, Hercules, CA, USA). Mouse anti-actin antibody (Santa Cruz Biotechnology) was used as a control of protein loading. Fiji ImageJ software was used to perform densitometry analysis. An equal area was selected inside each band, and the mean of gray levels (in a 0–256 scale) was calculated. Data were then normalized to values of background and of control actin band. The expression of α5 integrin was further confirmed by immunofluorescence analyses, as described above. Moreover, OPN expression was also evaluated by real-time PCR analysis. Briefly, total RNA was isolated from cells using the Quick-RNA MiniPrep kit (Zymo Research, Irvine, CA, USA), and reverse transcribed using the iScript cDNA synthesis kit (Bio-Rad Laboratories). The mRNA expression was measured by real-time PCR in a CFX96 Touch Real-Time PCR system (Bio-Rad Laboratories). A pre-validated set of PrimePCR™ SYBR^®^ Green Assay (Bio-Rad Laboratories) was used to quantify OPN (qHsaCID0012060) and GADPH (qHsaCED0037454) as a reference gene. Each reaction was performed in triplicate, and data were reported as expression relative to the GADPH gene (fold-change variation).

### 2.7. Evaluation of Inflammatory Cytokines in Cell Culture Media

In order to assess the OPN-secreted form involvement in inflammation, the presence of the main inflammatory cytokines, such as tumor necrosis factor (TNF) -α, interferon (IFN)-γ, and interleukin (IL)-1β, was assayed in supernatants of BSBs-treated hDPSCs, in co-culture with aPBMCs, by means of specific ELISA kits, according to the manufacturer’s instructions (RayBiotech, GA, USA). In particular, after incubating each sample for 2.5 h at room temperature, with gentle shaking and multiple wash steps, the prepared biotinylated antibody was added to each well, and incubated for 1 h at the same temperature and shaking conditions. Other wash steps were then required, before the addition of the provided streptavidin solution to each well. After 1 h of incubation, the prepared TMB one-step substrate reagent was added to each well and incubated for 1 h. At the end of the assay, 50 ul of stop solution was added to each well, and immediately read at 450 nm using iMark^TM^ Microplate reader (168-1135, Bio-Rad Laboratories).

### 2.8. Evaluation of Osteogenic Differentiation

In order to evaluate how osteogenic differentiation was influenced by BSBs treatment, cells were seeded at 3 × 10^4^ cells/cm^2^. After cell adhesion, the standard culture medium was replaced with the osteogenic medium (α-MEM, 10% FBS, 2 mM L-glutamine, 100 U/mL penicillin, 100 mg/mL di streptomycin, 100 nM dexamethasone, and 10 mM β-glycerophosphate, all from Sigma-Aldrich), added with 10 mg/mL of each BSB. After 10 days of induction, cell pellets and supernatants were collected for the evaluation of alkaline phosphatase (ALP) activity and bone metabolism, respectively. Firstly, ALP activity was measured with the alkaline phosphatase assay kit (ScienCell Research Laboratories, Carlsbad, CA, USA), according to the manufacturer’s protocol. Briefly, cell pellets obtained from undifferentiated and differentiated hDPSCs, cultured with and without BSBs, were lysed on ice for 20 min in lysis buffer. After brief sonication and microcentrifugation at 13,000 rpm for 15 min at 4 °C, the supernatants were collected and total protein concentrations were determined by Bradford assay (Sigma-Aldrich). To prepare samples, 90 μg of total protein was mixed with ALP assay buffer and transferred into a 96-well plate to reach the amount of 45 μL/well. Then, 5 μL of substrate was added to each well, and the plate was incubated at 25 °C in the dark for 60 min. To stop the reaction, 50 μL of stop solution was added to each well, and the absorbance was measured on iMark^TM^ Microplate reader (168-1135, Bio-Rad Laboratories), with a test wavelength at 405 nm and a reference wavelength at 630 nm. Data were represented by subtracting the 630 nm reference absorbance from the 405 nm measurement. Three replicates were prepared for each condition. Supernatants were analyzed using human bone metabolism antibody array (Abcam), according to manufacturers’ instructions. Concisely, after a first blocking step of 30 min, 100 μL of each sample was added to each well, and incubated overnight at 4 °C. Then, wash steps were performed before and after the incubation with detection antibody cocktail overnight at 4 °C, and another incubation with dye-conjugated streptavidin was carried out in the dark for 1 h at room temperature. After different wash steps, water droplets were completely removed by centrifugation, and signals were visualized by Tecan PowerScanner^TM^ (Thermo Fisher Scientific).

### 2.9. Statistical Analysis

All the experiments were performed in triplicate. Data were expressed as mean ± SD. Differences among three or more experimental samples were analyzed by ANOVA, followed by Newman–Keuls post hoc test (GraphPad Prism Software version 5 Inc., San Diego, CA, USA). In any case, significance was set at *p* < 0.05.

## 3. Results

### 3.1. Analysis of the Effects Induced by BSBs on hDPSCs Morphology and Stemness Properties

In order to assess the effects induced by BSBs, a morphological analysis on hDPSCs was performed at three different time points (Figure 1). As revealed by phase contrasts images, stem cells cultured alone maintain their fibroblast-like morphology throughout the whole culture time. The hDPSCs morphological alteration is observed when BSBs are added to culture media, triggering a prompt morphological alteration towards a more spherical one right after the treatment (0 h; high magnification square, Figure 1). It is noteworthy that hDPSCs fibroblast-like morphology is restored after 24 and 48 h of culture, suggesting that BSBs immediately affect hDPSCs adhesion without inducing a permanent effect.

The immuno-labelling with phalloidin, showing the organization of actin filaments, reveals that the hDPSCs fibroblast-like morphology is similar to the control group after 48 h of BSBs treatments (Figure 2A).

The maintenance of stem cell phenotype following BSBs treatments is investigated in immuno-selected STRO-1^+^/c-Kit^+^ hDPSCs, through the analysis of two typical mesenchymal/stromal stem cell marker expressions. As reported in Figure 2B, hDPSCs maintain the expression of c-Kit and STRO-1 after 48 h of BSBs treatments and, as revealed by high-magnification images (red squares), the expression of both stem cell receptors is strictly localized on the cell membrane. Based on these results, BSB treatments do not induce any alteration of stemness markers expression in hDPSCs after 48 h of culture.

### 3.2. Investigation of Stem Cell Adhesion, Proliferation and Viability after BSBs Treatments

Stem cell adhesion and proliferation are investigated by a real-time assay on attached cells treated with BSBs after 24 and 48 h, respectively. As reported in Figure 3A, after 24 h of treatment, a statistically significant reduction in stem cell adhesion is observed in all BSB-treated groups when compared to the control group (** *p <* 0.01, *** *p <* 0.001 vs. hDPSCs alone). Interestingly, a statistically significant reduction in stem cell adhesion is also observed in GTO-treated hDPSCs, when compared to both Gen-Os and Apatos-treated hDPSCs (^§^
*p <* 0.01 vs. hDPSCs + Gen-Os, hDPSCs + Apatos). After 48 h of treatment, a statistically significant decrease in stem cell proliferation is revealed in GTO-treated hDPSCs, when compared to both the control group and Gen-Os/Apatos-treated hDPSCs (** *p <* 0.01 vs. hDPSCs alone; ^§^
*p <* 0.05 vs. hDPSCs + Gen-Os, hDPSCs + Apatos). Moreover, the BSBs effects on hDPSCs growth is assessed inside each experimental group after 48 h. As shown in Figure 3B, the proliferating cells curve reveals the same increasing growth trend in each group from 24 to 48 h of treatment. These data highlight that these biomaterials affect stem cell adhesion rather than stem cell proliferation. The BSBs effect on stem cell proliferation is further supported by the analysis of PCNA, a marker for cells in early G1 phase and S phase of the cell cycle (Figure 3C). The densitometric analysis shows no statistically significant differences among the experimental groups, confirming the hypothesis that these biomaterials do not affect stem cell proliferation. Furthermore, a cell viability assay is performed at 24 and 48 h to evaluate the capability of BSB treatment to induce cell death (Figure 3D). Histograms reveal no statistically significant differences in terms of cell death in both 24 h and 48 h-treated groups, compared to the control group (black bars, Figure 3D). Taken together, these data suggest that BSB treatment induces a hDPSCs quick detachment without affecting cell viability, but causes a delay in terms of cell proliferation. Based on these results, 48 h of culture is the only time point considered for the subsequent experimental analyses.

### 3.3. Evaluation of Biological Effects of BSBs Treatment on hDPSCs

In light of these findings, revealing a prompt morphological alteration of BSB-treated hDPSCs gradually restored after 48 h, and a strictly related delay in cell proliferation, a deeper investigation on adhesion molecules expression was carried out. To this regard, the expression of integrin αVβ1, a crucial adhesion molecule that mediates the adherence of many cell types to components of the extracellular matrix, as well as cell–cell adhesion phenomena [25], was investigated in hDPSCs following BSB treatment. As shown in Figure 4A, no statistically significant differences of either integrin is observed among the experimental groups. Integrin αV expression is further confirmed by immunofluorescence analysis (Figure 4B).

In parallel, analysis of the expression of lamin A/C, a nuclear protein exerting many cell functions, including cell structural stability, is evaluated [26]. As reported in Figure 4C, Western blot analysis on BSB-treated hDPSCs shows no statistically significant differences among the experimental groups, even though a rising trend is observed in the lamin A/C expression (Figure 4C). Immunofluorescence analysis, reported in Figure 4C, shows the nuclear localization of lamin A/C as part of the nuclear lamina, a stiff meshwork consisting of different types of lamins localized between the nuclear envelope and chromatin [27]. Therefore, these results show that the quick detachment of hDPSCs following BSB treatment is not triggered by an alteration of adhesion molecules expression, further confirmed by the similar expression levels of lamin A/C, indicating cell structural stability.

Since BSBs are expected to possess bone regeneration properties, a preliminary evaluation of OPN expression is carried out on hDPSCs following BSB treatment. As shown in Figure 5A, real-time PCR analysis reveals that OPN mRNA levels are higher in treated samples than in the control. At the same time, Western blot analysis performed on whole lysates of BSB-treated hDPSCs shows no statistical differences among the experimental groups, even though a reduction in OPN expression is observed only in Apatos-treated hDPSCs, compared to both the untreated sample and GTO and Gen-Os-treated samples. OPN is a protein that also exists extracellularly, as a soluble cytokine involved in a broad range of functions, including inflammation and immune modulation [28,29]. Thus, it is widely believed that OPN may have a dual role in both bone formation and inflammation processes [30,31]. Therefore, in order to evaluate the capability of hDPSCs to release OPN in cell culture medium, Western blot analysis is also performed on supernatants of each experimental group. As reported in Figure 5C, OPN-secreted form is observed in BSB-treated hDPSCs, as well as in the control group, without any statistically significant differences.

With the purpose of investigating OPN-secreted form involvement in inflammation, BSB-treated hDPSCs are exposed for 48 h to an inflammatory microenvironment, mimicked by the presence of aPBMCs. The analyses of culture media for the main inflammatory cytokines, such as TNFα, IFNγ, and IL-1β, reveals the absence of these cytokines in supernatants (data not shown). These findings support the hypothesis that OPN produced by hDPSCs is involved in osteogenic differentiation, rather than inflammation.

### 3.4. Osteogenic Differentiation of hDPSCs Treated with BSBs

Given the role of OPN in the bone formation process, the ability of BSB treatment to affect hDPSCs osteogenic differentiation is further investigated. To this purpose, BSB-treated hDPSCs are cultured under osteogenic differentiation medium for 10 days. The osteogenic commitment is evaluated by means of ALP enzymatic activity (Figure 6A). Histograms show that, after 10 days of induction, hDPSCs undergo the osteogenic commitment, as revealed by the statistically significant higher levels of ALP activity, compared to hDPSCs undifferentiated (*** *p <* 0.001 vs. hDPSCs undiff). Moreover, a statistically significant increase in ALP activity levels is observed in GTO-treated cells, compared to the differentiated control group (^§^
*p <* 0.05 vs. hDPSCs diff). On the contrary, ALP activity levels are statistically significantly lower in Apatos-treated hDPSCs, compared to the same control group (^§§§^
*p <* 0.001 vs. hDPSCs diff). These findings suggest that BSBs treatment directly affects the osteogenic properties of hDPSCs.

In order to investigate how BSBs treatment influences the surrounding microenvironment, by exerting indirect effects on hDPSCs behavior, a bone metabolism assay is carried out on cell culture media (Figure 6B). Regarding bone morphogenic proteins (BMPs)-2, BMP-7, and OPN levels, histograms show a statistically significant decrease in all treated supernatants, with respect to the control group (^#^
*p <* 0.05, ^###^
*p <* 0.001 vs. Sup hDPSCs diff). Concerning the TGFβ3 expression, histograms reveal a statistically significant reduction in GTO/Gen-Os samples, compared to the control group (^##^
*p <* 0.01, ^###^
*p <* 0.001 vs. Sup hDPSCs diff). In parallel, a statistically significant decrease in PDGF-BB expression is observed only in the Gen-Os sample (^##^
*p <* 0.01 vs. Sup hDPSCs diff). Moreover, the detected MMP-3 levels in cell culture media are statistically significant lower in all treated supernatants (^#^
*p <* 0.05, ^###^
*p <* 0.001 vs. Sup hDPSCs diff). The quantitative assay also reveals a statistically significant increase in osteoprotegerin (OPG) in the GTO-supernatant, whereas a reduction in its levels is observed in the Apatos sample (^#^
*p <* 0.05, ^##^
*p <* 0.01 vs. Sup hDPSCs diff). Lastly, the RANKL and DKK-1 targets are evaluated. The analysis reveals that only Apatos treatment induces a statistically significant increase in both proteins in cell culture media (^#^
*p <* 0.05 vs. Sup hDPSCs diff). Taken together, these data confirm the hypothesis that these biomaterials modulate the secretion of the main bone metabolism-related cytokines, leading to an alteration in the surrounding microenvironment.

## 4. Discussion

Biomaterials in dentistry have been used for tissue engineering, for the development of novel strategies aimed at regenerating/repairing either soft or hard tissue defects. The grafting biomaterials are classified according to their origin in autologous, allograft, or xenograft bone. Although several disadvantages are documented, autologous bone still represents the gold standard treatment [11,12,13,14]. The use of xenografts represents a promising alternative that may overcome autografts limits, as well simplifying clinical procedures. These biomaterials are expected to be biocompatible, to trigger a limited activation of immune response, and to support the bone regeneration process [16,32,33]. These BSBs are applicable to multiple purposes, being differently formulated in order to fit critical, i.e., maxillary sinus augmentation (lateral or crestal), and non-critical defects, such as alveolar preservation [34].

It is noteworthy that the bone regeneration process is articulated in three consequential phases: inflammation, vascularization, and bone formation [35]. Physiologically, in bone defects, the damaged site is filled by a blood clot that is replaced by granulation tissue within 1 week [36]. Histologically, the granulation tissue is composed of infiltrate inflammatory cells in the loose connective tissue and new thin-walled capillaries [37], thus, inflammation and neoangiogenesis represent key processes that orchestrate the subsequent bone regeneration process [38,39,40]. During this regenerative process, osteoprogenitor cells (i.e., neural crest stem cells), osteoblasts, and osteoclasts represent the main cell types involved. These cells play in concert in a continuous cycle called bone remodeling [35,41]. In the present study, we evaluated the biological properties of human neural crest-derived stem cells obtained from dental pulp (hDPSCs) when exposed to three commercially available BSBs. Our study confirms the biocompatibility of BSBs, by evaluating cell morphology, adhesion, and proliferation of hDPSCs. It is of note that BSBs induce a rapid detachment of the cultured cells right after their first contact. After 24 h, hDPSCs recover their adhesion ability, and restore their spindle-shaped morphology. Through immunofluorescence and Western blot analysis, we confirm that, along the culture time, BSBs do not significantly alter the expression of either adhesion or structural stability molecules, such as integrin αV/β1 and lamin A/C, respectively. At the same time, the stemness phenotype of hDPSCs is maintained. Our hypothesis, underlying the mechanistic reasons of the rapid cell detachment induced by the BSBs contact, is represented by the rapid augmentation of calcium levels in cell media, induced by the solubilization of calcium phosphate released from the biomaterials. Interestingly, intracellular levels of OPN increase in hDPSCs when cultured with OsteoBiol^®^ GTO^®^ and OsteoBiol^®^ Gen-Os^®^. This increase might influence the inflammatory microenvironment, as well as contribute to the osteogenic commitment. Indeed, osteopontin is a multifunctional protein, mainly associated with bone metabolism, and also functions as a proinflammatory cytokine that is able to modulate the immune response [42,43,44]. Our data highlight that such increases do not relate to the inflammatory modulation (data not shown), but reflect a pro-anabolic function in bone metabolism, as revealed by increased ALP activity in pre-differentiated hDPSCs cultured with OsteoBiol^®^ GTO^®^ and OsteoBiol^®^ Gen-Os^®^. On the other hand, OsteoBiol^®^ Apatos^®^ decreases the ALP activity, suggesting its marginal role in osteogenesis. The paracrine effects exerted by hDPSCs cultured with BSBs are evaluated by a bone metabolism assay, in order to confirm how these biomaterials affect the bone remodeling process led by hDPSCs. Physiologically, bone remodeling involves the removal of mineralized bone by osteoclasts, followed by the activity of osteoblasts in formation of an osteoid matrix that subsequently becomes mineralized [45]. In this regard, our data highlight that the culture of hDPSCs with all the three BSBs reflects in a balanced secretion of factors that coordinate the inhibition of osteoblast activity (i.e., BMP-2, BMP-6, BMP-7, OPN, TGF-B3, and PDGF-BB) downstream in a blockade of the osteoclast activity (i.e., MMP-3, OPG, RANKL, and DKK1).

## 5. Conclusions

Taking all these considerations together, it is assumed that OsteoBiol^®^ GTO^®^ and OsteoBiol^®^ Gen-Os^®^ biomaterials exert mechanical functions and a osteogenic-promoting effect, and their clinical use in dentistry can be applied according to their commercial formulation. On the other hand, OsteoBiol^®^ Apatos^®^ does not exert pro-osteogenic properties and might be more appropriate as a bone void filler for critical size defects, since it exerts only a mechanical function.

## Figures and Tables

**Figure 1 polymers-14-02223-f001:**
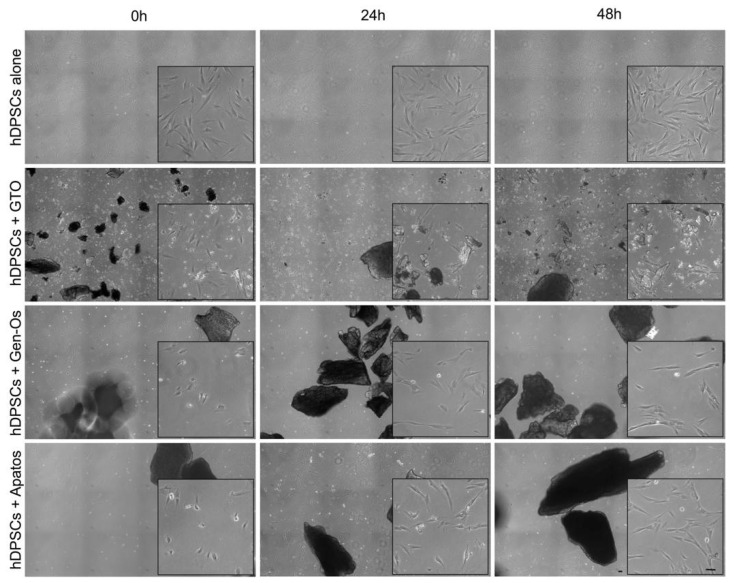
Morphological analysis of hDPSCs after BSBs treatment. Representative phase contrast grab images showing hDPSCs adhesion and morphology after 0, 24, and 48 h of BSBs treatment. High-magnification images are reported in black squares. Scale bar: 20 µm.

**Figure 2 polymers-14-02223-f002:**
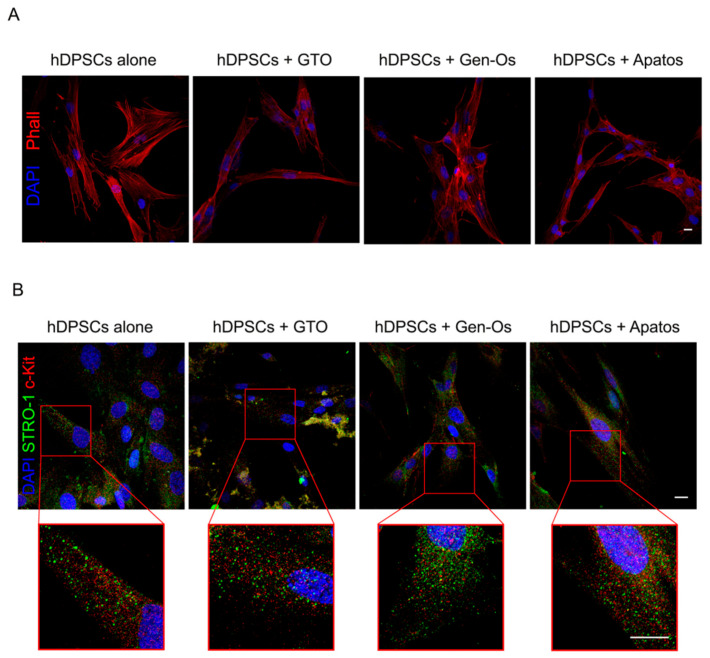
Evaluation of phalloidin and stemness markers on immunoselected hDPSCs. (**A**) hDPSCs morphology are evaluated by immunofluorescence analysis of phalloidin (PHA)-stained cells, after 48 h of BSBs treatment. Nuclei are counterstained with DAPI. Scale bar = 10 μm. (**B**) Evaluation by immunofluorescence analysis of stemness markers STRO-1 (green) and c-Kit (red), after 48 h of BSBs treatment. High-magnification images showing markers localization are reported in red squares. Nuclei are counterstained with DAPI. Scale bar = 10 µm.

**Figure 3 polymers-14-02223-f003:**
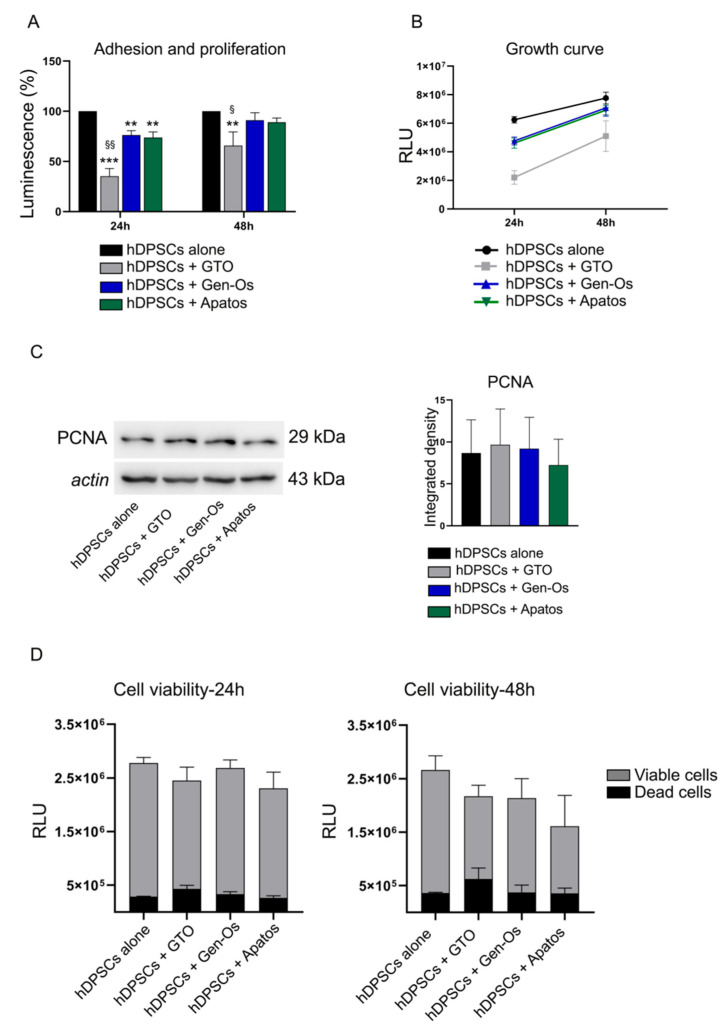
Stem cell adhesion, proliferation, and viability after BSB treatments. *(***A**) Cell adhesion (24 h) and proliferation (48 h) of attached hDPSCs treated with BSBs. Histograms report a statistically significant reduction in stem cell adhesion in all BSB-treated groups, compared to the control group (** *p <* 0.01, *** *p <* 0.001 vs. hDPSCs alone). GTO-treated hDPSCs show a statistically significant reduction in both stem cell adhesion and proliferation, when compared to either control group or Gen-Os and Apatos-treated groups (** *p <* 0.01 vs. hDPSCs alone; ^§^
*p <* 0.05, ^§§^
*p <* 0.01 vs. hDPSCs + Gen-Os, hDPSCs + Apatos). Data are reported as % of luminescence ± SD. (**B**) Growth curve assesses the effects of each BSB on stem cell growth after 48 h of treatment. The proliferating cells curve reveals the same increasing growth trend in each group from 24 to 48 h of treatment. Data are reported as relative luminescence unit (RLU) ± SD. (**C**) Western blot analysis of PCNA in hDPSCs treated with BSBs for 48 h. The densitometric analysis shows no statistically significant differences among the experimental groups. (**D**) Stem cell viability assay after 24 h and 48 h of BSB treatment. Histograms reveal no statistically significant differences in terms of cell death in both 24 h and 48 h-treated groups, compared to the control group (black bars). Data are reported as RLU ± SD.

**Figure 4 polymers-14-02223-f004:**
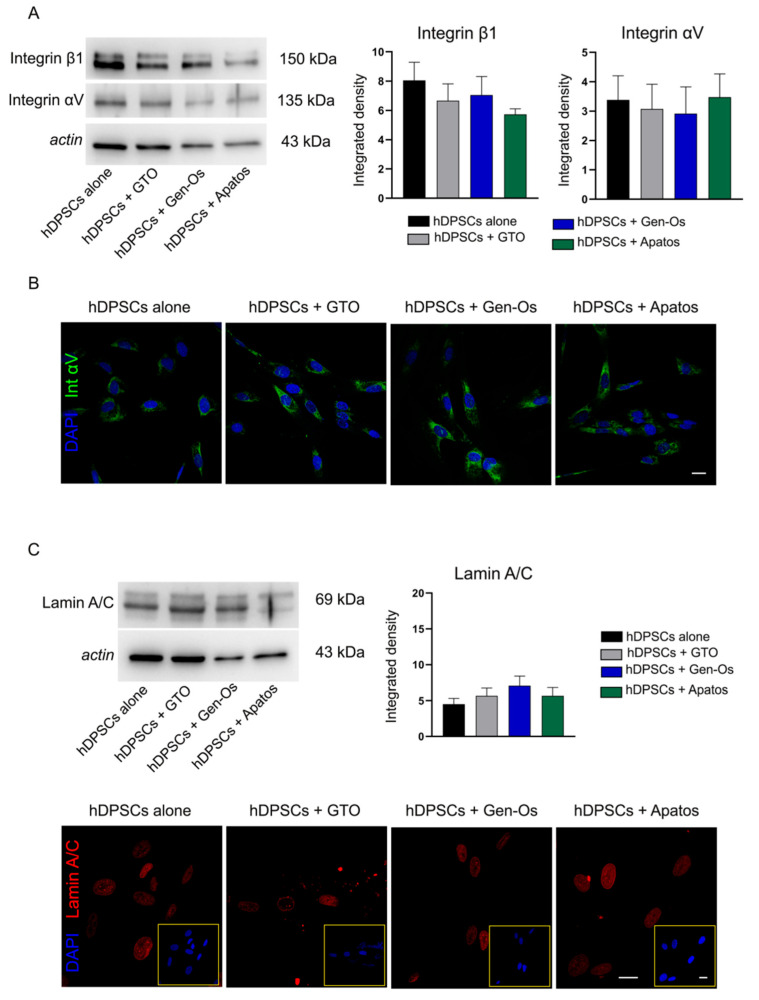
Evaluation of integrin αV/β1 and lamin A/C expression. (**A**) The expression of integrin αV and β1 in hDPSCs following 48 h of BSB treatment is evaluated using Western blot analysis. The densitometric analysis reveals no statistically significant differences in either integrin among the experimental groups. (**B**) Confocal immunofluorescence analysis of integrin αV on hDPSCs treated with BSBs for 48 h. Nuclei are counterstained with DAPI. Scale bar = 10 µm. (**C**) Western blot analysis of lamin A/C reveals no statistically significant differences among the experimental groups. The perinuclear localization of lamin A/C is confirmed by immunofluorescence analysis. Yellow squares show nuclei stained with DAPI. Scale bar = 10 µm.

**Figure 5 polymers-14-02223-f005:**
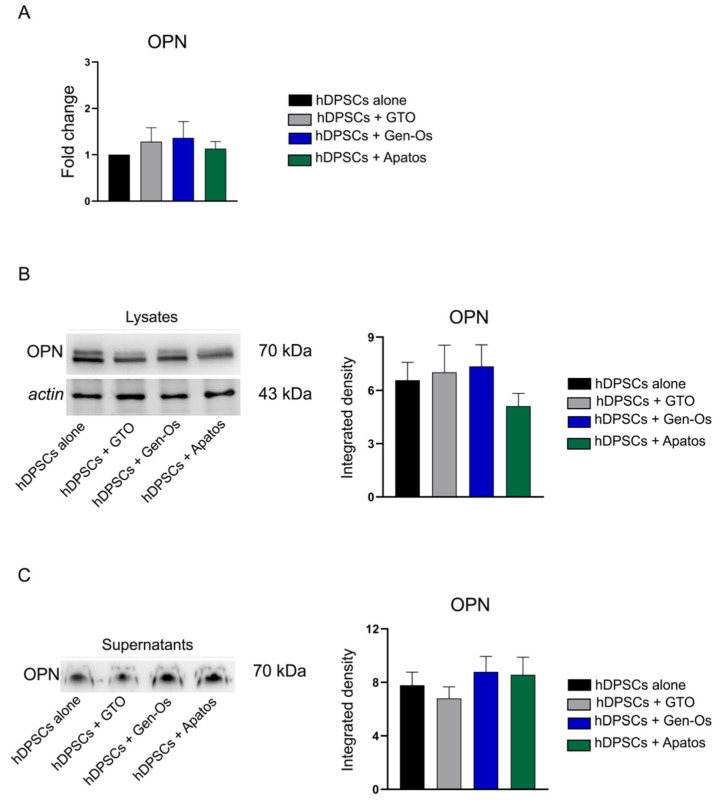
Assessment of OPN expression in hDPSCs following BSB treatment. (**A**) Evaluation of OPN expression by real-time PCR analysis reveals that OPN mRNA levels are higher in treated samples than hDPSCs alone. (**B**) Western blot analysis of OPN performed on whole lysates of BSB-treated hDPSCs shows no statistical differences among the experimental groups. (**C**) The release of OPN in cell culture media by hDPSCs is detected through Western blot analysis on supernatants of each experimental group (50µL). Histograms show that OPN-secreted form is observed in BSB-treated hDPSCs, as well as in the control group, without any statistically significant differences.

**Figure 6 polymers-14-02223-f006:**
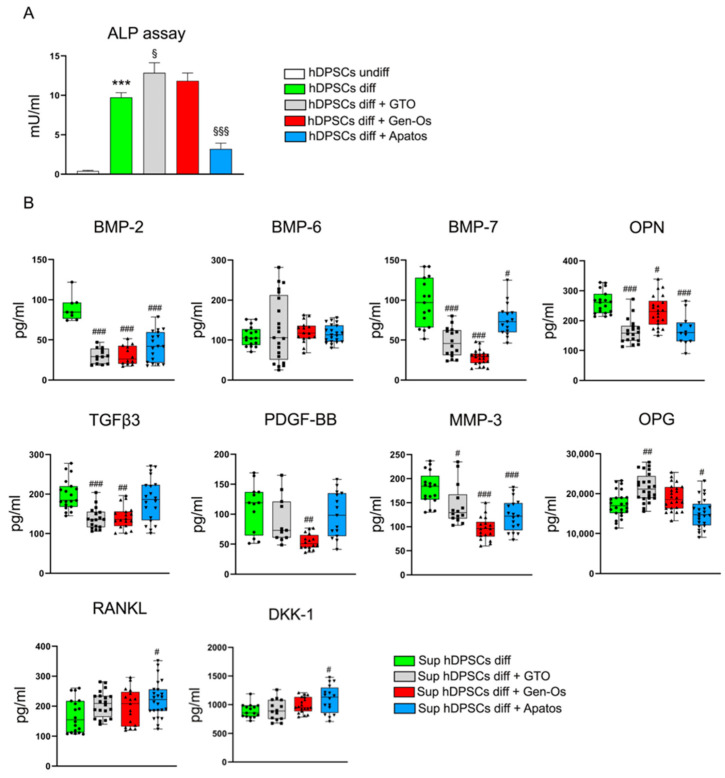
Osteogenic differentiation of hDPSCs treated with BSBs. BSB-treated hDPSCs are cultured under osteogenic differentiation medium for 10 days. (**A**) The osteogenic commitment is evaluated by means of ALP enzymatic activity. Histograms show that, after 10 days of induction, hDPSCs undergo the osteogenic commitment, as revealed by the statistically significant higher levels in ALP activity, compared to hDPSCs undifferentiated (*** *p <* 0.001 vs. hDPSCs undiff). The effects of BSBs on hDPSCs osteogenic differentiation are highlighted by statistically significant increased ALP activity levels in GTO-treated cells, and lower ALP activity levels in Apatos-treated hDPSCs, both compared to the differentiated control group (^§^
*p <* 0.05, ^§§§^
*p <* 0.001 vs. hDPSCs diff). (**B**) Bone metabolism assay is performed on cell culture media of the differentiated hDPSCs following BSBs treatment. Any statistically significant differences of each target are evaluated by comparing the treated samples to the control group (^#^
*p <* 0.05, ^##^
*p <* 0.01, ^###^
*p <* 0.001 vs. Sup hDPSCs diff).

## Data Availability

The data presented in this study are available within the article.

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
