# Peer review of "Characterization of Dental Pulp Stem Cells Response to Bone Substitutes Biomaterials in Dentistry"

_polymers, 2022, doi:10.3390/polym14112223_

Round 1

Reviewer 1 Report

There is considerable wording and grammatical errors throughout. Please review carefully.

Why the BSBs were added to culture medium at the concentration of 10 mg/mL? Is this a optimized concentration?

Coculture of BSBs and hDPSCs only last 48 hours. Why such a short time window? 

What's the purpose of co-culture PBMCs with hDPSCs? I didn't see any results related to culure of PBMCs.

The main inflammatory cytokines were assayed in supernatants of 198 BSBs-treated-hDPSCs in-co-culture with aPBMCs by means of specific ELISA kits. This is just the proteins that cells secreted into the supernatants. Did the authors consider digest the cells to check the intracellular proteins?

Author Response

  1. Why the BSBs were added to culture medium at the concentration of 10 mg/mL? Is this a optimized concentration?

RE: We thank the reviewer for the comment. The working concentration of 10 mg/mL was determined following preliminary evaluations of the effects of different concentrations of BSBs on culture parameters such as variations of culture medium pH. In particular, one of the used BSBs, i.e. Osteobiol GTO, still at low concentrations, induced a variation of pH. We evaluated different dosages of this BSB and the following  pH values were measured after 24 hours of culture:

  • 100 mg/ml: pH = 6.9
  • 70 mg/ml: pH = 7.08
  • 50 mg/ml: pH = 7.18
  • 20 mg/ml: pH = 7.2
  • 10 mg/ml: pH = 7.4
  • 5 mg/ml: pH = 7.4
  • 0 mg/ml (culture medium alone): pH = 7.4

Based on these preliminary results we used the first working concentration able to maintain physiological pH values and, accordingly, the same working concentration was set for the other BSBs, i.e. Osteobiol Apatos and Gen-Os.  

  1. Coculture of BSBs and hDPSCs only last 48 hours. Why such a short time window? 

RE: we thank the reviewer for the comment. Our experimental evaluations in terms of cell adhesion, proliferation, morphology and viability  were set in order to appreciate eventual differences among the different experimental groups after early time of treatment with BSBs. Indeed, at later times of culture, we expect a flattened trend in cultured cells which would be not affected anymore by the BSBs treatment. 

  1. What's the purpose of co-culture PBMCs with hDPSCs? I didn't see any results related to culure of PBMCs.

RE: we thank the reviewer for the observation. The co-culture between hDPSCs and pre-activated PBMCs in presence of BSBs was carried out to evaluate the biocompatibility properties of BSBs. Clinically, the BSBs are used for different dentistry procedures aimed to fit critical and non-critical defects. As well established in literature, the biomaterials used in this field are expected to be biocompatible, to trigger a limited activation of immune response and to support the bone regeneration process. Thus, one of the aims of this study was to evaluate the mechanisms underlying the co-culture system between hDPSCs and pre-activated PBMCs that resemble the inflammatory microenvironment. The results obtained did not show any statistically significant differences suggesting that BSBs did not affect  the co-culture environment. 

  1. The main inflammatory cytokines were assayed in supernatants of BSBs-treated-hDPSCs in-co-culture with aPBMCs by means of specific ELISA kits. This is just the proteins that cells secreted into the supernatants. Did the authors consider digest the cells to check the intracellular proteins?

RE: we thank the reviewer for the observation. In general, an inflammatory response is reflected in a huge secretion of cytokines in culture medium, that in this study mimics the microenvironment. To this purpose, all the cell secreted factors should be properly quantified through a highly sensitive assay, such as ELISA. The investigation of intracellular proteins involved in inflammatory processes did not fall within the aims of our study.

Reviewer 2 Report

Dear Tinco et al.,

The manuscript “Characterization of Dental Pulp Stem Cells response to Bone Substitutes Biomaterials in dentistry” (polymers-1742395) by Tinco et al. indicate that i) hDPSCs proved to be a suitable stem cell source to study the effects of BSBs and that ii) the formulation of BSBs may condition the biological properties of stem cells suggesting their versatile suitability to different dentistry applications. The topic is interesting, but I think this article should reconsider after proper changes in major revision for publication in Polymers. Some of my specific comments are below:

  1. In the abstract section (line 17-30), the authors should add quantitative results rather than only qualitative results.
  2. Describe the novelty of the article made by the author? From the results of my evaluation, it seems that many similar published works adequately explain what you have raised in the current manuscript related to Dental Pulp Stem Cell. If there are something others really new in this manuscript, please highlight it more clearly in the introduction section (line 33-74).
  3. The state of the art and the significance of the current study are not clearly present, the authors should highlight it more advanced in the introduction section (line 33-74).
  4. In the introduction section (line 33-74), the authors should explain the previous research conducted and its shortcomings. It will uphold the research gap that you filled with your research novelty. I recommend the authors elaborate on their introduction section. Do not forget to attention carefully my previous comments on numbers 2 and 3.
  5. To enrich the references related to biocompability of biomaterials, the authors are encourage and advice to adopt some specific literature published by MDPI as follow:
    • Tresca Stress Simulation of Metal-on-Metal Total Hip Arthroplasty during Normal Walking Activity. Materials (Basel). 2021, 14, 7554. https://doi.org/10.3390/ma14247554
    • The Effect of Bottom Profile Dimples on the Femoral Head on Wear in Metal-on-Metal Total Hip Arthroplasty. Journal of Functional Biomaterials. 2021, 12, 38. https://doi.org/10.3390/jfb12020038
  1. In the materials and methods section (line 76-245), the authors should add one systematic figure to illustrate the workflow of experimental testing in the present study to make the reader more interested and easier to understand rather than only using dominant text to explain.
  2. The author must provide a detailed specification and use condition more detail regarding all tools used in the research carried out so that the reader can estimate the accuracy and differences in the results that the authors describe due to the use of different tools in future studies.
  3. In the Results section (line 247-444), the authors are advised to compare the results they obtain with previous similar/identical studies if it is possible.
  4. The authors should add of one paragraph about the limitations of the presented study.
  5. The conclusion section is missing and should be included.
  6. In the whole of the manuscript, the authors sometimes made a paragraph only consisting of one or two sentences that made the explanation not clearly understood. The authors need to extend their explanation to become a more comprehensive paragraph. In one paragraph, it is recommended to consist of at least 3 sentences with 1 sentence as the main sentence and the other sentences as supporting sentences. For example in line 55-56.
  7. I see some errors on English in some areas of the present manuscript. To improve the quality of English used in this manuscript and make sure English language, grammar, punctuation, spelling, and overall style are correct, further proofreading is needed. As an alternative, the authors can use the MDPI English proofreading service for this issue.
  8. Please make sure the authors have used the Polymers, MDPI format correctly. The authors can download published manuscripts by Polymers, MDPI, and compare them with the present author's manuscript to ensure typesetting is appropriate. For example:
    • In line 1, type of the paper should be written as “Article”, not “Research Article”
    • All of the author's email should use black color without underline (line 6-16)
    • Uppercase and lowercase for title (line 2-3) and keywords (line 31) is not appropriate
    • And other

I am pleased to have been able to review the author's present manuscript. Hopefully, the author can revise the current manuscript as well as possible so that it becomes even better. Good luck for the author's work and effort.

Best regards,

The Reviewer

Author Response

The manuscript “Characterization of Dental Pulp Stem Cells response to Bone Substitutes Biomaterials in dentistry” (polymers-1742395) by Di Tinco et al. indicate that i) hDPSCs proved to be a suitable stem cell source to study the effects of BSBs and that ii) the formulation of BSBs may condition the biological properties of stem cells suggesting their versatile suitability to different dentistry applications. The topic is interesting, but I think this article should reconsider after proper changes in major revision for publication in Polymers. Some of my specific comments are below:

  1. In the abstract section (line 17-30), the authors should add quantitative results rather than only qualitative results.

RE: we thank the reviewer for the suggestion, we did not include quantitative data in the abstract since it is aimed to provide a brief summary of the objectives and the key results of the study, which find the proper elucidation across the main text and related sections. 

  1. Describe the novelty of the article made by the author? From the results of my evaluation, it seems that many similar published works adequately explain what you have raised in the current manuscript related to Dental Pulp Stem Cell. If there are something others really new in this manuscript, please highlight it more clearly in the introduction section (line 33-74).
  2. The state of the art and the significance of the current study are not clearly present, the authors should highlight it more advanced in the introduction section (line 33-74).
  3. In the introduction section (line 33-74), the authors should explain the previous research conducted and its shortcomings. It will uphold the research gap that you filled with your research novelty. I recommend the authors elaborate on their introduction section. Do not forget to attention carefully my previous comments on numbers 2 and 3.

RE: We thank the reviewer for the kind suggestion. According to the existing literature, the authors have highlighted the novelty of their study in the introduction section. As a matter of fact, to the best of our knowledge most of the studied biomaterials have been investigated from a clinical point of view. Our research has focused on the effects of BSBs on a suitable stem cell source by investigating the mechanisms underlying their use.

  1. To enrich the references related to biocompability of biomaterials, the authors are encourage and advice to adopt some specific literature published by MDPI as follow:
  • Tresca Stress Simulation of Metal-on-Metal Total Hip Arthroplasty during Normal Walking Activity. Materials (Basel). 2021, 14, 7554. https://doi.org/10.3390/ma14247554
  • The Effect of Bottom Profile Dimples on the Femoral Head on Wear in Metal-on-Metal Total Hip Arthroplasty. Journal of Functional Biomaterials. 2021, 12, 38. https://doi.org/10.3390/jfb12020038

RE: the bibliography section has been implemented as suggested by the reviewer. 

  1. In the materials and methods section (line 76-245), the authors should add one systematic figure to illustrate the workflow of experimental testing in the present study to make the reader more interested and easier to understand rather than only using dominant text to explain.

RE: we thank the reviewer for the wise suggestion. A schematic figure depicting the experimental design of the study has been included. 

  1. The author must provide a detailed specification and use condition more detail regarding all tools used in the research carried out so that the reader can estimate the accuracy and differences in the results that the authors describe due to the use of different tools in future studies.

RE: We thank the reviewer for the suggestion. In our study, standard technical procedures have been carried out, including immunofluorescence, western blot, real time PCR analyses and ELISA assay, according to well established protocols available in literature or provided by manufacturers. 

  1. In the Results section (line 247-444), the authors are advised to compare the results they obtain with previous similar/identical studies if it is possible.

RE: we thank the reviewer for the comment, however our results are highly innovative and cannot be compared with previous studies investigating Osteobiol BSBs effects.  

  1. The authors should add of one paragraph about the limitations of the presented study.

RE: we thank the reviewer for the comment. In light of our results and based on the fact that the studied BSBs are already available for dentistry clinical practice no negative effects were observed. Our mechanistic in vitro data further highlight their renowned clinical efficacy for the treatment of critical and non-critical defects.

  1. The conclusion section is missing and should be included.

RE: we agree with the reviewer's suggestion and a conclusion section has been added. 

  1. In the whole of the manuscript, the authors sometimes made a paragraph only consisting of one or two sentences that made the explanation not clearly understood. The authors need to extend their explanation to become a more comprehensive paragraph. In one paragraph, it is recommended to consist of at least 3 sentences with 1 sentence as the main sentence and the other sentences as supporting sentences. For example in line 55-56.

RE: we thank the reviewer for the suggestion and re-formatted our manuscript accordingly. 

  1. I see some errors on English in some areas of the present manuscript. To improve the quality of English used in this manuscript and make sure English language, grammar, punctuation, spelling, and overall style are correct, further proofreading is needed. As an alternative, the authors can use the MDPI English proofreading service for this issue. 

RE: we thank the reviewer for the kind suggestion. The manuscript was read and revised by an English native speaker. 

  1. Please make sure the authors have used the Polymers, MDPI format correctly. The authors can download published manuscripts by Polymers, MDPI, and compare them with the present author's manuscript to ensure typesetting is appropriate. For example:
  • In line 1, type of the paper should be written as “Article”, not “Research Article”
  • All of the author's email should use black color without underline (line 6-16)
  • Uppercase and lowercase for title (line 2-3) and keywords (line 31) is not appropriate
  • And other

RE: We thank the reviewer for the observation. We corrected the manuscript by using appropriate typesetting.

I am pleased to have been able to review the author's present manuscript. Hopefully, the author can revise the current manuscript as well as possible so that it becomes even better. Good luck for the author's work and effort.

Round 2

Reviewer 1 Report

I have no further questions and comments. I recomend to publish this paper.

Reviewer 2 Report

Dear Tinco et al.,

After carefully reading the author's revised manuscript entitled "Characterization of Dental Pulp Stem Cells response to Bone Substitutes Biomaterials in dentistry" (polymers-1742395) by Tinco et al., The authors have been made significant improvements in the revised manuscript. Also, all of the issue in my review report have been addressed precisely.

With my pleasure, I recommend the manuscript should be accepted for publication on Polymers.

Best regards,

The Reviewer